# Osteocytes Exposed to Titanium Particles Inhibit Osteoblastic Cell Differentiation via Connexin 43

**DOI:** 10.3390/ijms241310864

**Published:** 2023-06-29

**Authors:** Hao Chai, Qun Huang, Zixue Jiao, Shendong Wang, Chunguang Sun, Dechun Geng, Wei Xu

**Affiliations:** 1Department of Orthopedics, The Second Affiliated Hospital of Soochow University, Suzhou 215004, China; 2Department of Orthopedics, The First Affiliated Hospital of Soochow University, Suzhou 215004, China

**Keywords:** periprosthetic osteolysis, osteocyte, connexin 43, β-catenin

## Abstract

Periprosthetic osteolysis (PPO) induced by wear particles is the most severe complication of total joint replacement; however, the mechanism behind PPO remains elusive. Previous studies have shown that osteocytes play important roles in wear-particle-induced osteolysis. In this study, we investigated the effects of connexin 43 (Cx43) on the regulation of osteocyte-to-osteoblast differentiation. We established an in vivo murine model of calvarial osteolysis induced by titanium (Ti) particles. The osteolysis characteristic and osteogenesis markers in the *osteocyte-selective Cx43 (CKO)-deficient* and *wild-type (WT) mice* were observed. The calvarial osteolysis induced by Ti particles was partially attenuated in *CKO mice*. The expression of β-catenin and osteogenesis markers increased significantly in *CKO mice*. In vitro, the osteocytic cell line MLO-Y4 was treated with Ti particles. The co-culturing of MLO-Y4 cells with MC3T3-E1 osteoblastic cells was used to observe the effects of Ti-treated osteocytes on osteoblast differentiation. When Cx43 of MLO-Y4 cells was silenced or overexpressed, β-catenin was detected. Additionally, co-immunoprecipitation detection of Cx43 and β-catenin binding in MLO-Y4 cells and MC3T3-E1 cells was performed. Finally, β-catenin expression in MC3T3-E1 cells and osteoblast differentiation were evaluated after 18α-glycyrrhetinic acid (18α-GA) was used to block the intercellular communication of Cx43 between MLO-Y4 and MC3T3-E1 cells. Ti particles increased Cx43 expression and decreased β-catenin expression in MLO-Y4 cells. The silencing of Cx43 increased the β-catenin expression, and the over-expression of Cx43 decreased the β-catenin expression. In the co-culture model, Ti treatment of MLO-Y4 cells inhibited the osteoblastic differentiation of MC3T3-E1 cells and Cx43 silencing in MLO-Y4 cells attenuated the inhibitory effects on osteoblastic differentiation. With Cx43 silencing in the MLO-Y4 cells, the MC3T3-E1 cells, co-cultured alongside MLO-Y4, displayed decreased Cx43 expression, increased β-catenin expression, activation of Runx2, and promotion of osteoblastic differentiation in vitro co-culture. Finally, Cx43 expression was found to be negatively correlated to the activity of the Wnt signaling pathway, mostly through the Cx43 binding of β-catenin from its translocation to the nucleus. The results of our study suggest that Ti particles increased Cx43 expression in osteocytes and that osteocytes may participate in the regulation of osteoblast function via the Cx43 during PPO.

## 1. Introduction

Joint arthroplasty can relieve joint pain and enhance the quality of life of patients with severe osteoarthritis. The aseptic loosening of the prosthesis is the main postoperative complication of joint arthroplasty, and periprosthetic osteolysis (PPO)is the most important factor in aseptic loosening [1,2]. However, the precise mechanism underlying PPO remains unknown. The existing literature shows that friction during the movement of artificial joints induces the production of implant-derived particles, including titanium (Ti), chromium, and polyethylene, which exist at the interface between the prosthesis and host bone. The production of these particles triggers chronic inflammation and activates osteoclasts, leading to bone resorption. Meanwhile, these particles inhibit osteoblast function, decrease bone formation, and stimulate the expression of mediators that participate in the communication between osteoblasts and osteoclasts, thus increasing the ratio of nuclear factor-kappa B ligand (RANKL)/ osteoprotegerin (OPG)and further promoting bone resorption [3,4,5,6]. The delicate balance between bone formation and bone resorption is disturbed by wear debris, which results in unbalanced remodeling in favor of resorption [7].

Osteocytes are terminally differentiated osteoblasts that are deeply buried in the bone matrix and comprise 90% of all bone cells [8]. Accumulating evidence indicates that osteocytes play an important role in regulating bone formation and resorption. Osteocytes release RANKL to stimulate osteoclast differentiation and maturation and produce inflammatory cytokines to inhibit osteoblast differentiation [9,10]. The apoptosis of osteocytes increases the number of osteoclasts and promotes osteoclastic bone resorption [11,12]. Osteocytes also regulate osteoblastic bone formation through the canonical Wnt signaling pathway [13,14,15]. Some studies have shown that osteocytes also play an important role in PPO induced by wear debris. Lohmann [16] demonstrated that the addition of ultra-high-molecular-weight polyethylene (UHMWPE)to MLO-Y4 osteocytes in vitro significantly increased prostaglandin E2 and nitric oxide levels. Wang et al. demonstrated that a conditioned medium from osteocytes challenged with Ti-alloy particles promotes osteoclast formation [17]. Our study showed that osteolysis on the skull surface of mice increased in response to a high level of sclerostin, which is a characteristic product of osteocytes [18]. We further found that the treatment of MLO-Y4 cells with Ti particles inhibited MC3T3-E1 osteoblastic differentiation when the two types of cells were co-cultured via direct cell-to-cell contact [19]. Therefore, we speculated that connexin 43 (Cx43), which helps to perform gap junctional intercellular communication (GJIC), might play a certain role in the regulation of osteocyte-to-osteoblast differentiation.

Cx43 is the predominant connexin in skeletal tissues. A growing amount of research has focused on the role of Cx43 in the activity of osteocytes and osteoblasts [20]. Cx43 is closely associated with bone cell differentiation, skeletal metabolism, and bone remodeling, especially in mechanical environments [21,22,23]. However, the role of Cx43 in the regulation of osteocyte-to-osteoblast differentiation during PPO has not been conclusively determined.

In the current study, we built Ti-particle-induced osteolysis models in vivo and in vitro to investigate the effects of Cx43 on PPO. We found that Cx43 silencing attenuated particle-induced osteolysis. Additionally, Cx43 played an important role in the regulation of osteocyte over osteoblast during the osteolysis induced by wear debris.

## 2. Results

### 2.1. The Calvarial-Particle-Induced Osteolysis Was Partially Attenuated in Osteocyte-Selective Cx43-Deficient (CKO) Mice

To investigate the effects of Cx43 and osteocytes on PPO, we obtained transgenic mice with a conditional deletion of Cx43, specifically in osteocytes. We established a murine calvarial osteolysis model and found that the skull surfaces of mice from the WT group were smoother than those in the Ti group, indicating a higher degree of osteolysis in the Ti group. In contrast, the degree of osteolysis in the CKO + Ti group was significantly lower than that in the Ti group (Figure 1). Quantitative analysis revealed that, in comparison to the WT and Ti group, the Ti particles led to extensive lytic pores and decreased bone mineral density (BMD), bone volume/tissue volume (BV/TV), and trabecular number (Tb.N). In addition, trabecular separation (Tb.sp)was significantly higher in the Ti group than in the WT group in mice calvariae. Compared with the Ti group, the CKO + Ti group showed a reduction in the number of pores, an increase in BMD, BV/TV, and Tb.N, and a significant decrease in Tb.SP (Figure 1A).

### 2.2. The Inhibition of Osteoblastic Differentiation in Calvarial-Particle-Induced Osteolysis Was Partially Reversed in CKO Mice

To assess whether the alleviation of particle-induced osteolysis in the calvarial osteolysis model was associated with osteogenic differentiation, immunohistochemical staining was performed for osteogenic marker Osterix. The quantity of results that revealed the presence of Osterix-positive cells within the sagittal suture in the Ti group decreased significantly compared with the WT group. However, in the CKO + Ti group, the presence of Osterix-positive cells increased significantly in comparison with the Ti group (*p* < 0.05, Figure 1B).

### 2.3. The Protein Expression of β-Catenin, Runx2, Osterix, Alkaline Phosphatase (ALP), and Osteocalcin (OCN) Was Elevated in the Femur of CKO Mice

Immunohistochemical staining of femur bone sections from *CKO mice* showed that, compared with the WT group, the proportion of Cx43-positive cells in bone trabecula decreased significantly in the CKO group, whereas the proportion of β-catenin-positive cells increased significantly (*p* < 0.05, Figure 2A), which meant that β-catenin increased primarily in the osteocyte of *CKO mice*. Furthermore, we extracted proteins from the femurs of *CKO* and *WT mice*, and the results showed that the protein expression of β-catenin, Runx2, Osterix, ALP, and OCN increased in *CKO mice* compared to *WT mice* (*p* < 0.05, Figure 2B). The results of in vivo experiments showed that Cx43 deficiency in osteocytes activated the Wnt/β-catenin signaling pathway and promoted osteoblast differentiation.

### 2.4. Ti Particles Increased Cx43 Expression and Decreased β-catenin Expression in MLO-Y4 Cells

We examined the expression of Cx43 in MLO-Y4 cells exposed to various concentrations of Ti particles. Through immunofluorescent staining, we found that Cx43 expression increased alongside an increase in Ti particle concentration, which appeared green and was distributed along the cytomembrane of osteocytic cells (Figure 3A). The Western blot data further showed that Ti particles in the 0.1 and 0.2 mg/mL group at 24 h significantly increased the protein expression of Cx43 compared to the control (*p* < 0.05). The protein expression of β-catenin decreased significantly with an increase in Ti particle concentration compared with the level in the control group at 24 h (*p* < 0.05) (Figure 3B). The amount of β-catenin translocated into the nucleus decreased significantly with an increase in Ti particle concentration compared to the control (*p* < 0.05) (Figure 3C).

### 2.5. Cx43 Expression in MLO-Y4 Cells Reduced β-Catenin Expression

To verify the relationship between Cx43 and β-catenin found in vivo, β-catenin changes were detected after Cx43 silencing or overexpression in vitro. For Cx43 silencing, three interference sequences of Cx43–shRNA were designed and transfected into MLO-Y4 cells using a lentivirus vector. The Cx43–shRNA sequences were screened and identified. The first short hairpin RNA (shRNA-1) sequence significantly reduced the expression of Cx43, and therefore we chose shRNA-1 for subsequent experiments. Likewise, the overexpression of Cx43 was identified via Western blotting (Figure 4A). Free β-catenin must be translocated into the nucleus to activate the Wnt signaling pathway. To measure the level of nuclear β-catenin, we extracted nuclear protein and detected the level of nuclear β-catenin by Western blotting. Results showed that β-catenin levels in the MLO-Y4 nucleus increased significantly after Cx43 silencing (*p* < 0.05), and β-catenin levels decreased significantly when Cx43 was overexpressed (*p* < 0.05) (Figure 4B). These results suggested that Cx43 may negatively regulate the Wnt signaling pathway in osteocytic MLO-Y4 cell.

### 2.6. In the Trans-Well Co-Culture System, Cx43 Silencing of MLO-Y4 Reversed the Inhibition of Osteoblastic Differentiation of MC3T3-E1 Cells Induced by Exposure of the Co-Cultured MLO-Y4 Cells to Ti Particles

To ascertain the effects of MLO-Y4 cells exposed to Ti particles on MC3T3-E1 cells, a Millicell culture insert plate (Millipore, Billerica, MA, USA), which comprises a membrane perforated with multiple pores, was used to achieve the co-culturing of two kinds of cells, and two cells were easily separated for detection. MC3T3-E1 and MLO-Y4 cells were seeded on the basal and apical side of the membrane separately. After 7 and 14 days of osteo-induction, total protein of MC3T3-E1 was collected and the osteoblastic differentiation of MC3T3-E1 cells was evaluated. In the co-culture model, Ti particles, which only interacted with MLO-Y4, indirectly decreased the protein expression of β-catenin, Runx2, Osterix, ALP, and OCN in the MC3T3-E1 cells (*p* < 0.05, Figure 5A). Compared with the NC group, the ratio of the positive staining area to the total area of ALP and alizarin red staining in the Ti-treated group decreased significantly (*p* < 0.05, Figure 5B). 

After Cx43 silencing of MLO-Y4, the levels of some osteoblastic differentiation markers of MC3T3-E1 cells, such as β-catenin, Runx2, Osterix, ALP, and OCN, increased even when MLO-Y4 cells were treated with Ti particles. Likewise, the number of ALP-positive cells and mineralized nodules increased significantly after Cx43 silencing compared to the group treated with Ti particles (*p* < 0.05) (Figure 5A,B).

### 2.7. In the Co-Culture Model, Cx43 Silencing in the MLO-Y4 Cells Reduced the Expression of Cx43, Increased the Expression of β-Catenin Expression in MC3T3-E1 Cells, and Promoted Osteoblastic Differentiation

In the co-culture model, MLO-Y4 and MC3T3-E1 cells were in direct contact through the pores in the membrane of the insert plate. To explore expression changes of Cx43 on the interface of two kinds of cells, cells were seeded in a confocal small dish to form cell-to-cell contact randomly, and we induced osteogenic differentiation for seven days. Cx43 and F-actin were labelled green and red by immunofluorescence staining. Cx43 silencing in MLO-Y4 cells reduced the expression of Cx43 in MC3T3-E1 cells, whereas overexpression of Cx43 in MLO-Y4 cells increased the expression of Cx43 in MC3T3-E1 cells (Figure 6A).

To explore the effects of Cx43 expression changes on MC3T3-E1 cells in the co-culture model, osteoblastic differentiation of MC3T3-E1 cells was performed. The results showed that, when Cx43 was silenced in the MLO-Y4 cells, the protein expression of Cx43 in the MC3T3-E1 cells decreased and the protein expression of β-catenin, Runx2, Osterix, ALP, and OCN in the MC3T3-E1 cells increased significantly compared with the NC group (*p* < 0.05, Figure 6B). Colorimetric quantitative analysis of ALP staining and mineralized nodule staining showed that when Cx43 was silenced in the MLO-Y4 cells, more ALP-positive cells and more mineralized nodules were observed in the MC3T3-E1 cells than in the NC group (*p* < 0.05, Figure 6C). In contrast, the protein expression of Cx43 in MC3T3-E1 cells increased and the protein expression of β-catenin, Runx2, Osterix, ALP, and OCN in MC3T3-E1 cells decreased significantly compared with the NC group upon Cx43 overexpression in the MLO-Y4 cells (*p* < 0.05, Figure 6B). Fewer ALP-positive cells and mineralized nodules were observed in MC3T3-E1 cells than in the NC group when Cx43 was overexpressed in the MLO-Y4 cells (*p* < 0.05, Figure 6C).

### 2.8. Cx43 Negatively Regulated Osteoblastic Differentiation through the Inhibition of the Wnt/β-catenin Pathway in MC3T3-E1 Cells

As mentioned above, Cx43 silencing in MLO-Y4 cells induced low levels of Cx43 expression in MC3T3-E1 cells, increased β-catenin expression in MC3T3-E1 cells, and promoted osteoblastic differentiation in the co-culture model. To determine whether the effects of Cx43 expression change in MLO-Y4 cells on osteoblastic differentiation were associated with alterations in Cx43 expression in MC3T3-E1 cells, we evaluated the effects of Cx43 expression changes in MC3T3-E1 cells on osteoblastic differentiation. The results of immunofluorescent staining showed that, compared with the NC group, the expression of β-catenin and Runx2 increased when Cx43 was silenced in MC3T3-E1 cells. Conversely, the expression of β-catenin and Runx2 decreased when Cx43 was overexpressed in MC3T3-E1 cells (Figure 7A). When Cx43 was silenced in MC3T3-E1 cells, the protein expression of β-catenin, Runx2, Osterix, ALP, and OCN in MC3T3-E1 cells increased significantly compared with the NC group (*p* < 0.05, Figure 7B). When Cx43 was silenced in MC3T3-E1 cells, the protein expression of β-catenin in the nucleus increased significantly compared with the NC group (*p* < 0.05, Figure 7C). Likewise, when Cx43 expression was low, a higher number of ALP-positive cells and mineralized nodules were observed compared with the NC group (*p* < 0.05, Figure 7D). In contrast, the protein expression of β-catenin, Runx2, Osterix, ALP, and OCN decreased significantly compared with the NC group when Cx43 was overexpressed in MC3T3-E1 cells (*p* < 0.05, Figure 7B). The protein expression of β-catenin in the nucleus decreased significantly compared to the NC group when Cx43 was overexpressed in MC3T3-E1 cells (*p* < 0.05, Figure 7C). Fewer positive cells (cytosol blue coloration) and mineralized nodules were observed in MC3T3-E1 cells than in the NC group when Cx43 was overexpressed in the MLO-Y4 cells (*p* < 0.05, Figure 7D).

### 2.9. Cx43 Binding with β-Catenin Takes Part in Regulating the Wnt Signaling Pathway in MLO-Y4 and MC3T3-E1 Cells

Based on these results, we further investigated the association between Cx43 and β-catenin. We found that Cx43 (green) and β-catenin (red) could be co-located by labeling these two proteins using an immunofluorescence assay (Figure 8A). We further performed co-immunoprecipitation to demonstrate the binding of Cx43 and β-catenin, and the results showed that β-catenin was precipitated from the protein supernatant of MLO-Y4 and MC3T3-E1 cells using rabbit-anti-mouse Cx43 antibodies and magnetic beads (Figure 8B). Furthermore, Cx43 could also be precipitated from the protein supernatant of these 2 kinds of cell using rabbit-anti-mouse β-catenin antibodies and magnetic beads (Figure 8C). Subsequently, β-catenin and Cx43 were detected in the Cx43–/β-catenin–bead complex by Western blot analysis (Figure 8B,C).

### 2.10. In the Co-Culture Model, the Presence of Cx43–GJIC between MLO-Y4 Cells and MC3T3-E1 Cells Slightly Decreased the β-Catenin Expression of MC3T3-E1 Cells and Inhibited ALP Expression

In addition to being associated with the route by which signaling molecules lead to the regulation of intracellular signaling, Cx43 also participated in the constitution of gap junction and played an important role in cell-to-cell communication. We further investigated the role played by GJIC between these two kinds of cells. MLO-Y4 and MC3T3-E1 cells were co-cultured in confocal small dishes, as mentioned above, and 18-α glycyrrhetinic acid (18α-GA) was used to block Cx43–gap junctions between these two cells. β-catenin and F-actin were labelled green and red via immunofluorescence staining. Fluorescent staining results shows that both in the NC + 18α-GA group and Cx43-H + 18α-GA group, blocking Cx43–gap junctions slightly increased the expression of β-catenin in MC3T3-E1 cells. On the condition of 18α-GA intervention, β-catenin of MC3T3-E1 cells in Cx43-H group is always lower than that of the NC group (Figure 9A).

To explore the effects of blocking Cx43–gap junctions between these two cells on osteoblastic differentiation of MC3T3-E1, we performed ALP and alizarin red staining of osteoblasts in a direct cell-to-cell contact model. Colorimetric quantitative analysis of ALP staining indicated that blocking Cx43–gap junctions enhanced early osteoblast differentiation slightly. For mineralized nodule staining, blocking Cx43–gap junctions had no significant effect on the formation of calcium nodules (Figure 9B).

## 3. Discussion

As has been established, osteocyte bodies reside in the bone matrix of the lacuna, and their dendritic processes extend through tiny tunnels called canaliculi to adjacent osteocytes and other cells on the bone surface, directly or indirectly mediating intercellular communication [9]. This process plays an important role in bone remodeling [9,10,11,12,13,14]. PPO induced by wear particles causes an imbalance in bone remodeling. The balance between bone formation and bone resorption is disrupted, and the level of osteolysis is elevated. Emerging studies have shown that osteocytes play important roles in PPO by regulating bone formation and resorption [10,24]. Ormsby et al. [25] reported that human primary osteocyte-like cells displayed an upregulated expression of matrix metallopeptidase-13 (MMP13), carbonic anhydrase 2, and cathepsin K after exposure to both polyethylene and metal-wear particles. Our previous study showed that the inhibition of osteocytes by Ti particles further decreased osteoblast differentiation via direct cell-to-cell contact [19]. We speculated that direct intercellular communication played a certain role. It should be noted Cx43 is the most abundant gap junction protein expressed in bone cells and makes up gap junction and hemichannels. This plays an important role in cell-to-cell communication and the communication between bone cells and their extracellular microenvironment in the skeleton. In addition, Cx43 also serves as a scaffolding protein that is associated with signaling molecules, leading to the regulation of intracellular signaling independent of channel activity [26]. Therefore, the present study aimed to explore the effect of Cx43 on the regulation of osteocyte-to-osteoblast differentiation. 

In order to investigate the effect of Cx43, we established *conditioned Cx43 knockout mice*. Under the action of a mouse dentine matrix protein 1(Dmp1) (14.1 kb) promoter, the strain specifically expressed Cre recombinase in osteocytes but did so weakly in osteoblasts. To demonstrate the specificity of Cx43 knockout in osteocytes, we detected Cx43 expression in the samples of brain and muscle tissue by Western blotting. We found that there was no significant difference in Cx43 expression in these samples between *WT* and *CKO mice* (Appendix A). The mouse calvarial osteolysis model induced by Ti particles showed fewer lytic pores on the skulls of the *CKO mice* with selective Cx43 silencing than *WT mice*. Compared with the Ti group, the fewer lytic pores and increased BMD in the CKO + Tigroup may be associated with the enhanced osteogenic differentiation. The result of immunohistochemical staining for osteogenic marker Osterix supports this conjecture, namely, that the presence of Osterix-positive cells increased significantly in the CKO + Ti group compared with that in the Ti group. Likewise, Runx-2 and ALP increased significantly in the CKO + Ti group compared with that in the Ti group (Appendix A). The micro-CT results showed that we detected increased BMD, BV/TV, and Tb.Th, as well as decreased Tb.sp, in the distal femur of *CKO mice* compared with *WT mice* (Appendix A). The further results showed the protein expression of β-catenin, Runx2, Osterix, ALP, and OCN also increased in *CKO mice*. Runx2, a target protein of the Wnt signaling pathway, is an important transcription factor that promotes osteoblast differentiation [27]. Osterix, a downstream transcription factor of Runx2, reflects the ability of osteoblasts to differentiate early [28]. ALP, OCN, and mineralized nodules are markers of differentiation [27,28]. These results proved that Cx43 knockout activated the Wnt signaling pathway and promoted bone formation.

In the in vitro study, MLO-Y4 osteocytic cells and MC3T3-E1 osteoblastic cells were co-cultured using a Millicell culture insert plate, which comprises a membrane perforated with pores and allows dendritic processes of osteocytes to extend into contact with osteoblasts through the pores on the membrane. First, we found that Ti particles decreased β-catenin expression and nuclear translocation, and Ti particles increased Cx43 expression in MLO-Y4 cells. Cx43 silencing in MLO-Y4 promoted the osteoblastic differentiation of MC3T3-E1 cells and attenuated the inhibitory effects on the osteoblastic differentiation of MC3T3-E1 cells that were induced by the application of Ti treatment to the co-cultured MLO-Y4 cells. This finding corroborated the results of the in vivo study. Emerging evidence has ascribed unanticipated biological roles to connexins that go beyond direct intercellular communication, pointing towards broader functions of these membrane proteins, such as transcription, metabolism, autophagy, and ion channel trafficking [29,30,31]. Our study found low expressions of Cx43 in MLO-Y4 and MC3T3-E1 cells increased β-catenin expression and nuclear translocation, whereas overexpression of Cx43 decreased β-catenin expression; Cx43 negatively regulated β-catenin expression and inhibited the Wnt signaling pathway. Some studies have shown that the intracellular C-terminus of Cx43 is long and variable, and can bind to proteins in the cytoplasm. For example, some scholars observed that the binding of Cx43 and β-catenin influenced osteoblast differentiation [32]. However, the association between Cx43 and β-catenin in osteocyte has rarely been studied and it has not been determined as to whether this association is operative in wear-particle-induced osteolysis. Our study demonstrated that Cx43 and β-catenin co-located and bound together between osteocyte and osteoblast through immunofluorescence assay and co-immunoprecipitation. Increased Cx43 binding to β-catenin prevented β-catenin translocation to the nucleus, thus inhibiting the Wnt signaling pathway [33], which might explain why the inhibition of Cx43 attenuates the osteolysis induced by wear particles.

Additionally, we found that Ti particles increased Cx43 expression in osteocytes, and a Cx43 expression change in osteocytes induced synchronous changes in Cx43 levels in osteoblasts. When Cx43 was silenced in the MLO-Y4 cells, the level of Cx43 in the MC3T3-E1 cells was reduced. Conversely, Cx43 expression in the MC3T3-E1 cells increased with its overexpression in the MLO-Y4 cells. Some studies have shown that, on the surface of two adjacent membranes, only paired complete channels exist, whereas unpaired hemi-channels are digested by autophagy [30,31]. We speculated that MC3T3-E1 cells might correspondingly express more Cx43 to meet the increased level of Cx43 expression in the MLO-Y4 cells. During the challenging of osteocytes with Ti particles, the increase in Cx43 expression in both osteocytes and osteoblasts could mean that more gap junctions formed. In osteoblasts, increased Cx43 bound β-catenin and restricted β-catenin nuclear translocation. Therefore, there was limited free β-catenin nuclear translocation into the nucleus to activate Runx-2. In this way, the increased Cx43 of osteocytes indirectly leads to the inhibition of osteoblastic differentiation (Figure 10).

Cx43 silence affects the function of both the gap junction and hemichannels. Our previous study showed that osteocyte regulated osteoblast differentiation mostly via direct cell-to-cell contact, which meant gap junction was more important. To further study the effects of Cx43–gap junctional intercellular communication, we used 18α-GA to block the exchange of small molecules between MLO-Y4 and MC3T3-E1 cells with/without the overexpression of Cx43 in MLO-Y4 cells in vitro co-culturing. Blocking Cx43–GJIC slightly increased the expression of β-catenin and ALP stain in MC3T3-E1 cells. However, blocking Cx43–GJIC had no significant effect on the formation of calcium nodules. It might be that the side effects of 18α-GA affected the results, inhibiting the formation of calcium nodules [34]. Other possible reasons for the discrepancy may be differences in experimental conditions and different species of cells. For all this, the results showed that blocking Cx43–GJIC between MLO-Y4 and MC3T3-E1 cells partially promoted the Wnt/β-catenin signaling pathway and early osteoblast differentiation of MC3T3-E1 cells. Therefore, Cx43 acted as a docking platform, binding with β-catenin in MLO-Y4 and MC3T3-E1 cells to play a more significant role in osteoblast differentiation in comparison with Cx43–gap junctional intercellular communication. 

Some studies have also shown that Cx43 is a selective inhibitor of the Wnt signaling pathway. Xie et al. reported that Cx43 inhibited the canonical Wnt signaling pathway and the proliferation of osteosarcoma cells [35]. Bivi et al. reported that osteocytic MLO-Y4 cells with Cx43 knockdown exhibited higher β-catenin protein expression and enhanced response to mechanical stimulation [36]. Grimston reported that periosteal bone formation was activated in *Cx43-deficient mice* at a lower strain level compared to the *WT mice*, and also highlighted that trabecular bone mass was increased upon the addition of a load compared to the *WT mice* [37]. However, the results of studies focusing on Cx43 have been controversial. Loiselle et al. showed that Cx43 deficiency resulted in delayed osteoblastic differentiation and impaired restoration of biomechanical properties due to attenuated β-catenin expression relative to wild-type littermates [38]. The inhibition of Cx43 expression in osteoblastic cells results in the decreased expression of phenotypic characteristics of differentiated osteoblasts, including alkaline phosphatase, osteocalcin, and bone sialoprotein [39,40]. We speculated that the reasons behind these inconsistent findings might be due to different roles played by Cx43 in bone homeostasis, such as bone growth and development, fracture healing, and response to mechanical stress changes. Abnormal increases or decreases in Cx43 might not be good for the maintenance of bone homeostasis. In addition, the results of studies focusing on Cx43 might be different under different study and experimental conditions. For example, in terms of Cx43 regulating bone formation through the Wnt/β-catenin signaling axis [36,40,41], the resultant effects differ markedly between fracture healing and responses to mechanical loading. Furthermore, this might be associated with the kind of cell types used in a study [38,39,40].

Overall, our present study demonstrated that the exposure of osteocytes to Ti particles partially inhibited osteoblastic cell differentiation via Cx43. Additionally, the inhibition of osteocytic Cx43 attenuated the osteolysis induced by wear particles.

There are some limitations to this study. We studied the interaction between osteocytes and osteoblasts in vitro by using cell lines. Cell lines may not fully mimic the in vivo metabolism of cells. With the exception of Ti particles, other materials were not involved in our study, and the relevance of Cx43 and wear debris made up of other materials maybe need further study. We focused on exploring the interaction between Cx43 and β-catenin. The adenomatosis polyposis coli (APC)/Axin/ glycogen synthase kinase (GSK)/β-catenin complex has been called a “destruction complex” in Wnt signaling, while the role of the complex turnover on Cx43/β-catenin needs to be studied in the future. In addition to regulating bone formation, osteocytes also play a role in controlling bone resorption. Further studies on the regulation of osteocytes induced by wear debris on osteoclasts are needed.

## 4. Materials and Methods

### 4.1. Detection and Preparation of Ti Particle

Ti particles were obtained from the Nanjing Emperor Nano Materials Company. The sizes of the Ti particles ranged from 24.51 to 233.58 nm. These particles proved to be effective in our previous study [42].

The particles were prepared as previously described [18]. Endotoxin-free detection of Ti particles was performed and confirmed using a QCL-1000 kit (Biowhittaker, Walkersville, MD, USA). Ti particles were mixed with PBS at a concentration of 100 mg/mL, the stock solution was sonicated, and a concentration of 40 mg/mL was used to establish the calvarial osteolysis model. After being sonicated, the stock solution was diluted with a medium to 0.1 mg/mL or 0.2 mg/mL for the in vitro study.

### 4.2. Preparation of Conditioned Cx43 Knockout Mice

The global knockout of Cx43 is embryonically lethal. We utilized mice with the conditional deletion of the Cx43 encoding gene *Gap Junction Protein Alpha 1 (Gja1)* in the osteocytes. To obtain this kind of mice, Dmp-1-cre mice were used. Dmp1 is a marker of osteocytes. Driven by the DMP-1 promoter, the strain specifically expressed Cre-recombinase in odontoblasts and osteocyte, but weakly in osteoblasts. The mouse Dmp1 promoter we used is from reference, the length is about 14.1 kb. *H11*, located on *mouse chromosome 11*, is a safe site for foreign gene insertion. The foreign gene integrated into this site can be expressed stably and efficiently without destroying the function of endogenous genes. In this study, the Dmp1-iCre-PolyA gene fragment was inserted into the *H11* site of mice using Clustered Regularly Interspaced Short Palindromic Repeats-associated protein 9 (CRISPR/Cas9) technology. The brief process was as follows: the donor vector and sgRNA were constructed in vitro. Cas9, donor, and sgRNA (small guide RNA) were microinjected into the fertilized eggs of *C57BL/6J mice*, and *F0 generation mice* were obtained. The *F0-positive mice* were mated with *C57BL/6J mice* by PCR, sequencing, and Southern blot. Subsequently, the stable inheritance of *F1 positive mice model* was obtained. The strategy for generating transgenic mice is the conditional knockout of osteocytes. Knockout mice were provided by GemPharmatech Co. Ltd. Briefly, *Dmp1-cre mice* were mated with *Cx43 flox/wt mice* to obtain *Cx43 flox/wt (WT)* and *Dmp1-cre-Cx43 flox/wt mice*. The *Dmp1-cre-Cx43 flox/wt mice* were mated with *Cx43 flox/wt mice* to obtain *Dmp1-cre* and *Cx43flox/flox target mice*. Genotyping was performed by PCR using genomic DNA isolated from mouse earpieces and appropriate primers.

### 4.3. Establishment of a Ti-Particle-Induced Mouse Calvarial Osteolysis Model and Micro-CT Analysis

All animal experiments were approved by the Ethics Committee of the Soochow University (SUDA20210320A05, March 2021). Twenty male *C57BL/6 WT mice* and *conditioned Cx43 knockout mice (CKO)* at 6 weeks of age with an average weight between 20 and 25 g were used as subjects. The *WT mice* were divided into the NC group and the Ti-particle-treated (Ti) group, with 10 mice in both groups. Similarly, 20 conditioned *Cx43 knockout mice (CKO)* were divided equally into the CKO and CKO + Ti groups. All mice were anesthetized by intraperitoneal injection of pentobarbital sodium. The skin was incised and the periosteum was completely removed to expose the skull surface. In the Ti and CKO + Ti group, 40 μL of Ti particles (40 mg/mL) diluted with PBS was injected subcutaneously around the middle suture. Ti particles for injection were used as mentioned above in method Section 4.1. The mice in the NC and CKO groups were injected with 40 μL of PBS. Two weeks later, the mice were sacrificed for subsequent analysis using CO_2_. The cranial specimens (n = 5 per group) fixed with paraformaldehyde were analyzed with micro-CT (Scanoco) using 10 μm layers. Three micro-CT scan results of cranial specimens were selected for data analysis for each group (n = 3). The X-ray parameters were set to 70 kV and 114 μA. Micro-CT images were taken from a circular region of interest (ROI of 3 mm in radius) located at the middle of each calvaria, and data of BMD, BV/TV, Tb.N, Tb.Th, and Tb.Sp were collected. All the calvariae were collected for further analysis.

### 4.4. Hematoxylin and Eosin (HE) Staining and Immunohistochemical Staining

After incubation in formalin for two days, the calvariae of osteolysis model and femur specimens of *CKO* and *WT mice* (*n* = 5 per group) were decalcified in ethylenediaminetetraacetic acid (EDTA, Sigma-Aldrich, Burlington, MA, USA) for one month. The specimens were then trimmed and selected. Following dehydration and paraffin embedding, the calvariae were cut into 2 μm and the femur samples were cut into 5 μm sections for HE and immunohistochemical staining according to routine protocols. For immunohistochemical staining of Osterix of calvariaes and the Cx43 and β-catenin of femur specimens, paraffin sections were dewaxed with xylene and then hydrated with gradient ethanol. The calvariae tissues were incubated with the Osterix primary antibody (Abcam, Shanghai, China, 209484, 1:200 dilution). Femur specimens were incubated with Cx43 primary antibody (Abcam, 11039, 1:500 dilution) and β-catenin primary antibody (Proteintech, 66379, 1:200 dilution), which was followed by incubation with secondary antibodies. Diaminobenzidine (DAB) (Abcam, 64238) was used for staining the sections, along with counterstaining with hematoxylin, differentiation with hydrochloric acid ethanol, bluing with ammonia, and washing with double distilled water (DDW). Finally, the sections were dehydrated, cleared, and sealed separately. The sections were observed under a light microscope (Zeiss, Oberkochen, Germany). Three slides were selected from each group for quantitative analysis. In calvariae, Image J was used to count the proportion of positive cells (brown), which was calculated as follows: proportion of positive cells = (the number of positive cells in sagittal/the number of total cells in sagittal) × 100%. In femur specimens, the proportion of positive cells (brown) was calculated as follows:

proportion of positive cells = (the number of positive cells in cortical bone/the number of total cells in cortical bone) × 100%.


### 4.5. Extraction and Detection of Bone Tissue Protein from Cx43 Gene Knockout Mice

Femur specimens of *CKO* and *WT mice* (n = 5 per group) were soaked in 75% ethanol. Femurs were obtained from the mice by removing the soft tissue. The femur was pulverized and incubated with a lysis buffer. Protein samples were centrifuged, and the supernatants were used for subsequent analyses. The used protein separation, detection, and analysis methods are described below in the section describing the Western blot analysis. The expression of Cx43, β-catenin, Runx2, Osterix, ALP, and OCN proteins in the bone tissue was determined.

### 4.6. Cell Culture and Treatments

The osteocytic cell line MLO-Y4 and osteoblastic cell line MC3T3-E1 were obtained from the Chinese Academy of Sciences Cell Bank (Shanghai, China). MLO-Y4 cells were cultured in a modified essential medium (α-MEM) containing 10% fetal bovine serum (FBS, Gibco, Gaithersburg, MD, USA), as well as 1% penicillin and streptomycin. MC3T3-E1 cells were cultured in α-MEM with 10% Fetal bovine serum (FBS), 1% penicillin, and 1% streptomycin and in osteogenic differentiation medium consisting of 10% FBS, 10 mM β-glycerophosphate, and 50 μg/mL ascorbic acid (Sigma-Aldrich, St. Louis, MO, USA) for three days. The MLO-Y4 cells were separately incubated with Ti particles at concentrations of 0 (control), 0.1 mg/mL, and 0.2 mg/mL after 24 h of seeding, and fresh medium was supplied every three days. 

The model for co-culturing MLO-Y4 and MC3T3-E1 in vitro was constructed to evaluate the effects of Ti-treated osteocytes on osteoblastic differentiation using a Millicell culture insert plate (Millipore, Billerica, MA, USA), which comprises a membrane perforated with 1 μm pores, as previously described [18]. In the co-culture model, the insert plate was inverted, and the basal surface of the membrane was seeded with MC3T3-E1 cells at a density of 1 × 10^4^ cells/cm^2^ in a 500 μL basal medium and incubated for 2 h. The insert was then inserted into a Millicell 6-well tissue culture plate containing 1 mL of basal medium. The MLO-Y4 cells were seeded at a density of 2 × 10^3^ cells/cm^2^ on the apical side of the membrane (top side of the insert) containing 1 mL of basal medium and incubated overnight. After 24 h, the MLO-Y4 cells on the upper side were treated with a pure basal medium or Ti particles diluted with basal medium (Figure 11). The basal medium for the co-culture experiments consisted of α-MEM supplemented with 10% FBS, 1% penicillin, and 1% streptomycin. An osteogenic differentiation medium was used as described above after three days. In the co-culture model, the MC3T3-E1 cells were on the lower surface of the membrane to prevent them from contacting the Ti particles, thereby avoiding the potential direct effects of Ti particles on osteoblastic cells, and the cells were easily separated for the detection of osteoblast and osteocyte changes. The inhibition of gap junction function was accomplished by supplementing basal and osteogenic differentiation medium with 10 μM 18α-GA for the coculture period. Control groups were incubated with equivalent concentration of DMSO (0.1%).

### 4.7. Transfection for Cx43 Silencing or Overexpression

Cx43 was silenced using *short-hairpin shRNA* lentiviral particles. Briefly, after finding the sequence of the Cx43-encoding gene GJA-1 in the GenBank database, the interference sequence *GJA-1-shRNA* was designed by Jima Gene Co., Ltd. (Suzhou, China). The shRNA sequence was as follows: *5′-GGTGTCTCTCGCTCTGAATAT-3′*. The cells were infected with lentiviral particles carrying either scrambled or *GJA-1-specific shRNA*. The efficiency of the deletion was determined by Western blotting. Similarly, Cx43 overexpression lentivirus was obtained from Jima Gene Co., Ltd. The Cx43 overexpression cell line was established in the same manner as *GJA-1* silencing. 

### 4.8. Western Blot Analysis for Protein Expression

The protein expression of Cx43 and β-catenin in MLO-Y4 osteocytes is described in the following section. MLO-Y4 cells were seeded in 6-well plates and treated with different concentrations of Ti particles for 24 h. The cells were washed twice with PBS, treated with lysis buffer, placed on ice, and centrifuged. The supernatant was collected and the protein concentration was measured using a BCA protein assay kit (Beyotime, Shanghai, China, P0010). Approximately 30 μg of protein samples were separated by 10% SDS-PAGE and electro-blotted onto nitrocellulose membranes. After blocking with 5% bovine serum albumin (Sangon Biotech, Shanghai, China, 4240GR100), the membranes were incubated with a 1:1000 dilution of primary monoclonal antibodies against Cx43 (Abcam, Cambridge, UK, ab11370) and β-catenin (dephosphorylated form, CST, MA, USA, 8480T) overnight. After washing them four times with TBST (Tris-buffered saline with Tween), the membranes were incubated with horseradish peroxide (HRP)-conjugated goat anti-rat IgG (Multisciences, Hangzhou, China, GRT007) and goat anti-rabbit IgG (Multisciences, GAR007), and subsequently washed with TBST. Protein signals were illuminated with electrochemiluminescence (ECL) and analyzed using a GIS image analysis system.

The nucleoplasmic protein isolation kit (Thermo, Waltham, MA, USA, 89881) was used to detect β-catenin expression in the nucleus. Briefly, ice-cold CER I and CER II were added to the cells. After centrifugation, the supernatant was removed and the precipitate was centrifuged again, followed by ice lysis, which was repeated four times. Nuclear proteins in the supernatant were obtained, and β-catenin was detected using lamin-B1 as an internal reference. Protein analysis was performed as described above.

In the co-culture models using the Millicell Culture Insert Plate, the MC3T3-E1 cells were collected on day 7 for the detection of Cx43, β-catenin, ALP, Runx2, and Osterix, and on day 14 for the detection of OCN. MC3T3-E1 cells were collected using cell scrapers. Membranes were incubated with a 1:1000 dilution of primary antibodies against Cx43 (Abcam, ab11370), β-catenin, ALP (Abcam, 95462), Runx2 (Abcam, 76956), Osterix (Abcam, 209484), and OCN (Santa Cruz, CA, USA, 365797). Protein analysis was performed as previously described.

### 4.9. Immunofluorescence Staining

#### 4.9.1. Detection of Cx43 in MLO-Y4 Cells

MLO-Y4 cells were washed with PBS, fixed with 4% paraformaldehyde, permeabilized with 0.1% Triton X-100, sealed with 5% goat serum, and incubated with the Cx43 primary antibody diluted 1:500 with 2% goat serum. Then, goat anti-rabbit IgG H&L (Alexa Fluor 488) (ABCam, AB150077) was incubated and washed with phosphate-buffered saline tween (PBST)three times. The cells were then washed in PBST, and the nuclei were stained with DAPI (Beyotime, C1002). The stained cells were observed under a laser scanning confocal microscope LSM880 (Zeiss, Oberkochen, Germany) using an excitation wavelength of 488 nm.

#### 4.9.2. Simultaneous Detection of Cx43 and β-Catenin in MLO-Y4 and MC3T3-E1 Cells

MLO-Y4 cells were detected on day 2. To evaluate osteoblastic differentiation, MC3T3-E1 cells were examined after seven days of osteogenic induction. Cells were incubated with a mixture of β-catenin (Proteintech, Rosemont, IL, USA, 66379) and Cx43 antibodies overnight and then incubated with a mixture of donkey anti-mouse IgG H&L (Alexa Fluor^®^ 594) (Abcam, 150108) and goat anti-rabbit IgG H&L secondary antibodies. Finally, autofluorescence was quenched using the Auto Fluo Quencher C1212 kit. The cells were observed at excitation wavelengths of 488 and 594 nm.

#### 4.9.3. Detection of Cx43, β-Catenin in MC3T3-E1 Cells in the Co-Culture Models

For co-culture analysis, 2 × 10^3^ MLO-Y4 cells and 1.0 × 10^4^ MC3T3-E1 cells were co-inoculated in a confocal small dish and osteogenic differentiation was induced for seven days. The cells were first incubated with the Cx43 or β-catenin antibody overnight and then with the goat anti-rabbit IgG H&L (Alexa Fluor^®^ 647) (Abcam, 150083) secondary antibody. In order to establish a clear distinction in the morphological characteristics of these two types of cells, FITC phalloidin (Abcam, 176753) was added to stain the F-actin of these two kinds of cells. After quenching autofluorescence using the Auto Fluo Quencher C1212 kit, the cells were observed at an excitation wavelength of 647 and 488 nm. Images were collected as z-series and Z-series were performed with a z-step of 1.0 µm. (Zeiss, Oberkochen, Germany).

#### 4.9.4. Simultaneous Detection of β-Catenin and Runx2 in MC3T3-E1 Cells

MC3T3-E1 cells were detected after seven days of osteogenic induction. Cells were incubated with a mixture of β-catenin (CST, 8480T) and Runx2 (Abcam, 76956) primary antibodies and then incubated with a mixture of goat anti-rabbit IgG H&L (Alexa Fluor 488) and donkey anti-mouse IgG H&L (Alexa Fluor^®^ 594) secondary antibodies. After quenching autofluorescence using the Auto Fluo Quencher C1212 kit, the cells were observed at excitation wavelengths of 488 and 594 nm.

### 4.10. Co-Immunoprecipitation Detection of Cx43 and β-Catenin Binding in MLO-Y4 Cells and MC3T3-E1 Cells

MLO-Y4 and MC3T3-E1 cells were collected using a cell scraper, treated with Np-40 cell lysis buffer (Beyotime, Shanghai, China), and placed on ice, followed by centrifugation and collection of the supernatant. The protein concentration was detected and was adjusted to 1 μg/μL using Np-40 cell lysis buffer (Beyotime, Shanghai, China). Subsequently, 25 μL of protein G magnetic beads (Thermo, 01108614) were added for pre-cleaning. After incubation with gentle shaking, the magnetic beads were separated using a magnetic rack and the supernatant was collected for protein analysis. Rabbit IgG and anti-Cx43/β-catenin primary antibodies were then added to the protein followed by incubation. After this, 35 μL of magnetic beads were added with gentle shaking. The magnetic beads were collected using a magnetic rack and transferred to a new tube. Bound proteins were eluted with 2× protein-loading buffer after boiling. Cx43 and β-catenin were detected by Western blotting, as previously described.

### 4.11. ALP Staining and Mineralized Nodule Staining of MC3T3-E1 Cells

ALP staining was performed on day 7 and ALP levels were assessed by the 5-bromo-4-chloro-3-indoylphosphate and nitroblue tetrozolium (BCIP/NBT) color reaction using a BCIP/NBT ALP color development kit (Beyotime, C3206). Briefly, cells were fixed with 4% paraformaldehyde, and after washing three times, BCIP/NBT was added. In the co-culture models, the contralateral MLO-Y4 cells were removed and dried. The results were observed and quantified. The particle analysis tool of Image J was used to calculate the ratio of the stained area.

The mineralization of MC3T3-E1 cells was evaluated on day 14 after MLO-Y4 treatment with or without Ti particles, visualized with alizarin red staining (Cyagen, Santa Clara, CA, USA, S0141) and washed twice to remove the stain. In the co-culture models, the contralateral MLO-Y4 cells were removed and dried at room temperature. The results were observed and quantified. The particle analysis tool of Image J was used to calculate the ratio of the stained area.

### 4.12. Statistical Analysis

The data were analyzed using SPSS version 17.0. All data were expressed as the mean ± standard deviation, and each assay was independently repeated three times. Differences in statistical analyses were evaluated using one-way analysis of variance (ANOVA) and post hoc multiple comparisons. Differences were considered statistically significant at *p* < 0.05.

## 5. Conclusions

Cx43 in osteocytes negatively regulated osteoblastic differentiation, which is related to the suppression of the Wnt/β-catenin signal pathway in osteoblasts. Taken together, these findings indicate that osteocytes may participate in the regulation of osteoblast function via the Cx43 during PPO.

## Figures and Tables

**Figure 1 ijms-24-10864-f001:**
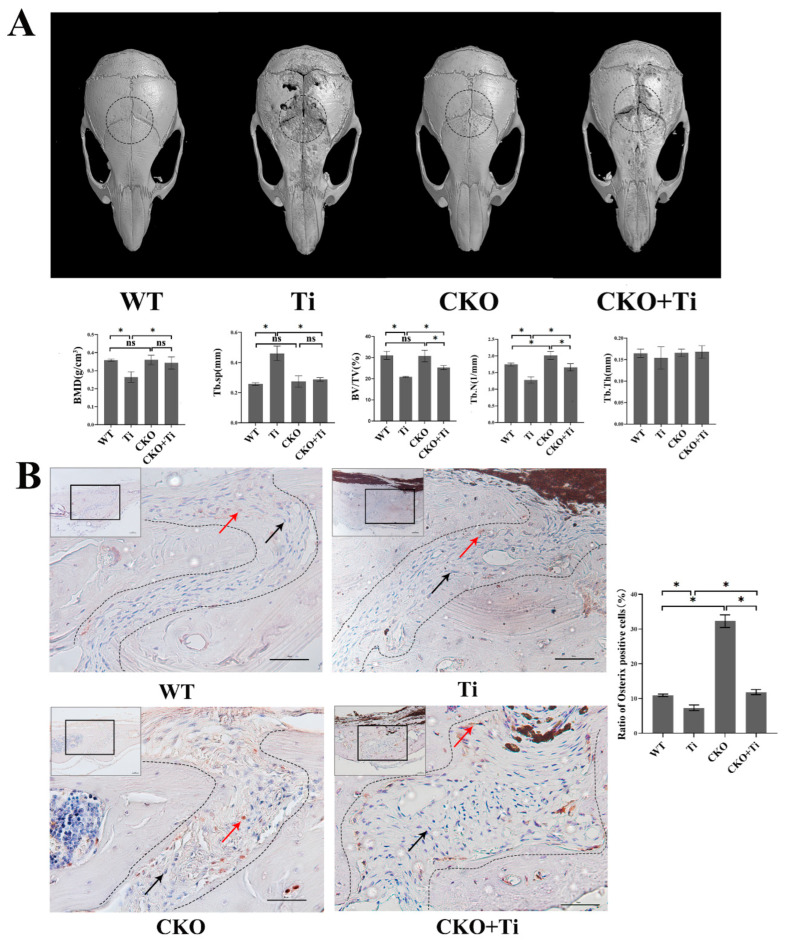
Partial attenuation of the calvarial-Ti-particle-induced osteolysis in *CKO mice*. (**A**) Mouse calvarial osteolysis samples were scanned using micro-CT. The dashed black circle indicated the ROI for data analysis. Three-dimensional images were generated and some parameters such as BMD, BV/TV, Tb.N, trabecular thickness (Tb.Th), and Tb.SP were obtained and analyzed using software CTVol (Version 3.1, Bruker)based on micro-CT scanning data. (WT: *wild-type mice* treated with phosphate-buffered saline (PBS); Ti: *wild-type mice* treated with 40 mg/mL of Ti particles; CKO: *Cx43-deficient mice* treated with PBS; CKO + Ti: *Cx43-deficient mice* treated with 40 mg/mL of Ti particles). Three micro-CT scan results of cranial specimens were selected for data analysis for each group. (**B**) Immunohistochemical stain of Osterix. Representative sections were chosen from the sagittal exploration of mouse calvaria. The black boxed regions were viewed at a higher magnification level. The areas within the curves indicate the sagittal suture. Red arrows represent Osterix-positive cells (brown), and black arrows represent Osterix-negative cells (blue) in the sagittal suture. The scale bar is 50 μm. Statistic results are displayed on the right on the basis of the statistical results quantification of the Osterix-positive cell proportion in the sagittal of mouse calvarial. Three slides were selected from each group for quantitative analysis. The numbers of Osterix-positive cells and total cells within the sagittal suture were counted using the Image J software (Image J 1.8.0; National Institutes of Health, USA). The proportion of positive cells (brown) was calculated as follows: the proportion of positive cells = (the number of positive cells in sagittal/the number of total cells in sagittal) × 100%. Data are expressed as mean ± SD, n = 3, * *p* < 0.05, NS: no significant.

**Figure 2 ijms-24-10864-f002:**
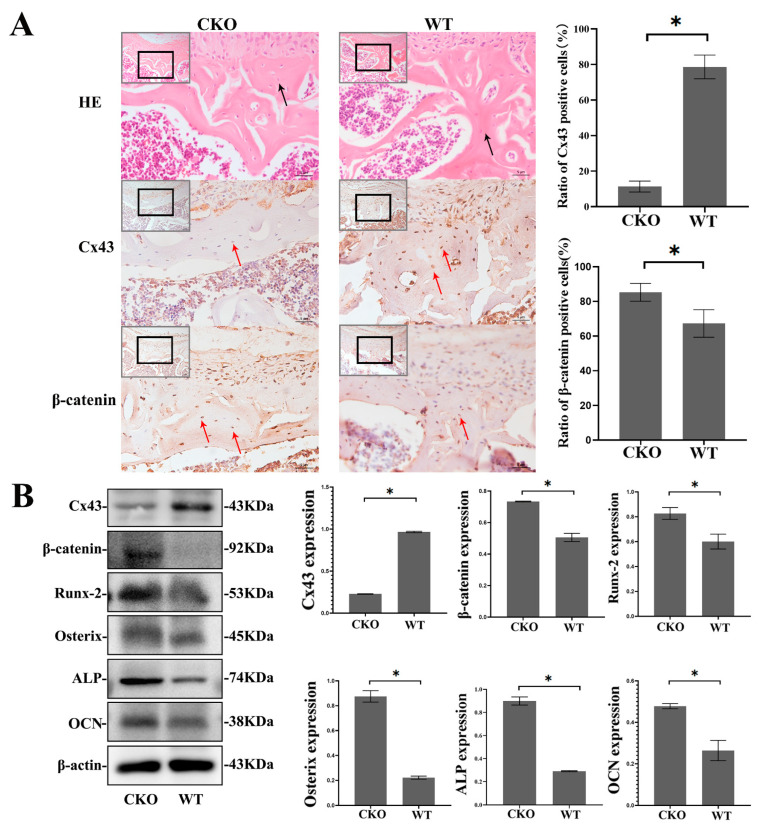
Increased protein expression of β-catenin, Runx2, Osterix, ALP, and OCN in the femurs of CKO mice. (**A**) Representative sections of HE and immunohistochemical staining showed were chosen from femoral trochlear groove. Osteocytes were located in cortical bones of femurs of *WT* and *CKO mice* (black arrows). Immunohistochemical staining of β-catenin and quantification of Cx43 and β-catenin-positive cells. Red arrows indicate Cx43 and β-catenin immunopositive cells (brown). Bar graph shows the proportion of Cx43 and β-catenin-positive cells. (The images were magnified 20 times in gray boxes and 40 times in black boxes separately. The scale bar is 5 μm). Three slides were selected from each group for quantitative analysis. Numbers of immunopositive cells and total cells in cortical bone were counted using the Image J software (Image J 1.8.0; National Institutes of Health, USA). In femur specimens, the proportion of immunopositive cells (brown) was calculated as follows: proportion of positive cells = (the number of positive cells in cortical bone/ the number of total cells in cortical bone) × 100%. (**B**) The proteins from the femurs of mice were extracted and the protein expression levels of β-catenin, Runx2, Osterix, ALP and OCN were determined. (*CKO: Cx43-deficient mice*, *WT: wild type mice*). Data are expressed as mean ± SD, n = 3, * *p* < 0.05.

**Figure 3 ijms-24-10864-f003:**
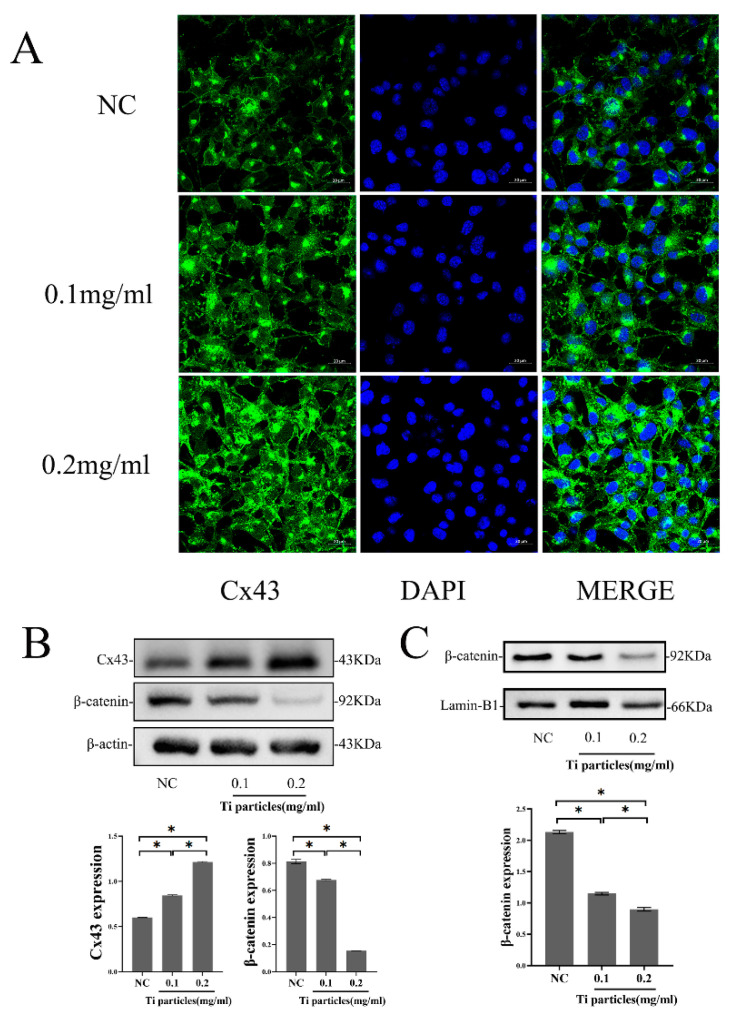
Ti particles increased Cx43 expression and decreased β-catenin expression in MLO-Y4 cells. (**A**) Immunofluorescence showed the distribution of Cx43 in MLO-Y4 cells treated with different concentrations of Ti particles for 24 h (Cx43: green; DAPI: Blue Scale: 20 μm). (**B**) Western blot analysis of the protein expression of β-catenin and Cx43 in MLO-Y4 cells exposed to different concentrations of Ti particles for 24 h. (**C**) Western blot analysis of the β-catenin in the nucleus of MLO-Y4 cells exposed to different concentrations of Ti particles for 24 h. Lamin-B1 was selected as an internal reference for nuclear protein. (NC group; 0.1 mg/mL, 0.2 mg/mL: the concentrations of Ti particles). Data are expressed as mean ± SD, n = 3, * *p* < 0.05.

**Figure 4 ijms-24-10864-f004:**
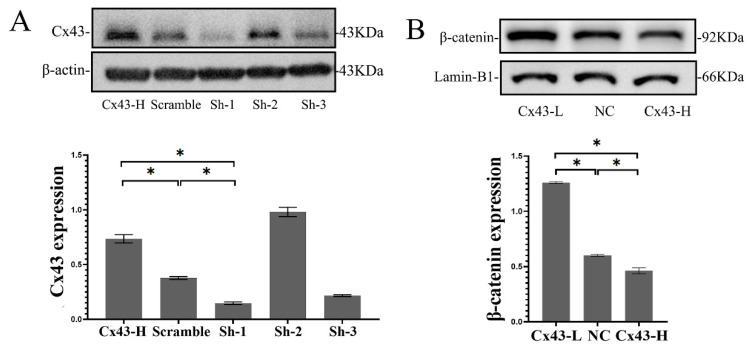
Cx43 reduced β-catenin expression in MLO-Y4 cells. (**A**) Cx43 silencing or overexpression was detected by Western blotting. (**B**) The β-catenin in the nucleus of MLO-Y4 cell was detected by Western blotting and Lamin-B1 was selected as an internal reference for nuclear protein. (Cx43-L: the group with low Cx43 expression; Cx43-H: the group with overexpression of Cx43; NC: negative control group). Data are expressed as mean ± SD, n = 3, * *p* < 0.05.

**Figure 5 ijms-24-10864-f005:**
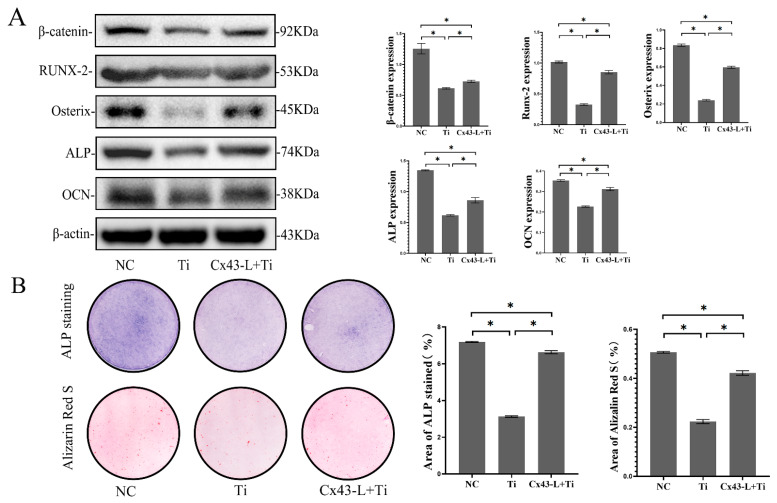
MLO-Y4 cells exposed to Ti particles inhibited the osteoblastic differentiation of MC3T3-E1 cells in the co-culture model, and Cx43 silencing attenuated the inhibition effects of Ti-induced MLO-Y4 on osteoblastic differentiation of MC3T3-E1. In the co-culture model, MLO-Y4 cells were treated with 0.2 mg/mL of Ti particles. (**A**) Protein expression of β-catenin, Runx2, Osterix, ALP, and OCN in MC3T3-E1 cells was detected by Western blotting. (**B**) ALP and alizarin red staining of MC3T3-E1 cells on days 7 and 14 were evaluated separately. ALP active site was dark blue or blue-purple and calcium nodules were stained as red. (NC: negative control group; Ti: group of MLO-Y4 cells exposed to Ti particles; Cx43-L+Ti: group of MLO-Y4 cells with low expression of Cx43 exposed to Ti particles) The particle analysis tool of Image J was used to calculate the ratio of the stained area. Data are expressed as mean ± SD, n = 3, * *p* < 0.05.

**Figure 6 ijms-24-10864-f006:**
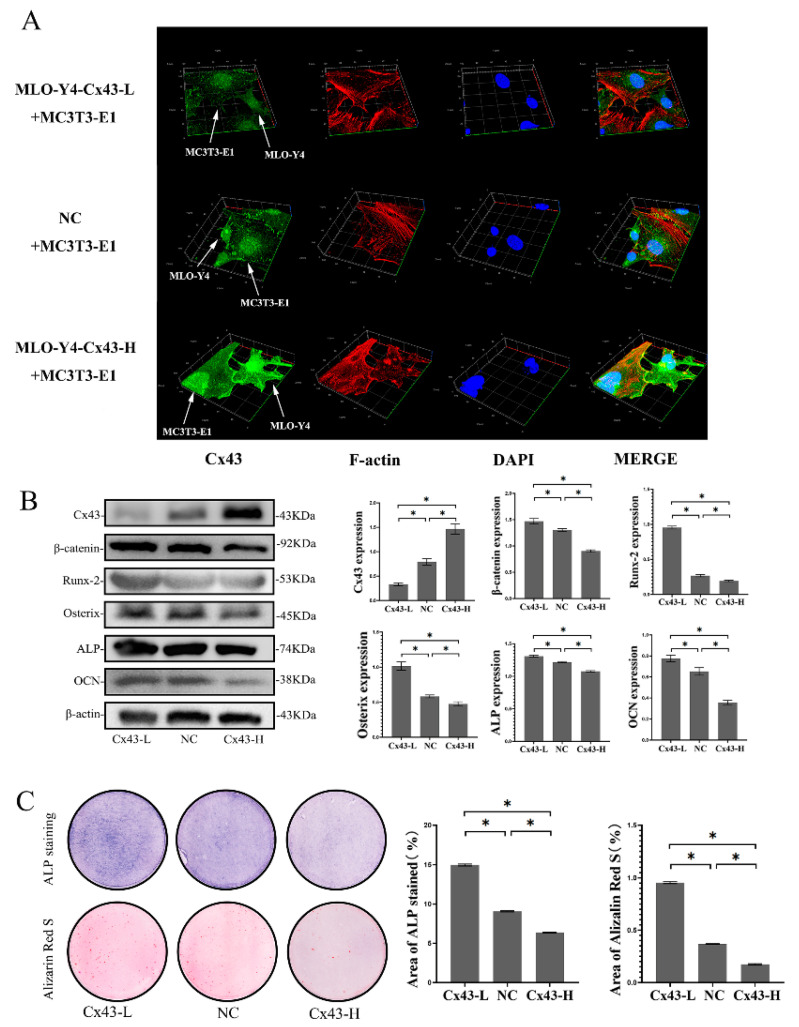
Cx43 silencing in MLO-Y4 cells reduced the expression of Cx43 in MC3T3-E1 cells, increased β-catenin expression in MC3T3-E1 cells and promoted osteoblastic differentiation in the co-culture model. (**A**) Immunofluorescent staining results after 7 days of co-culture (MLO-Y4-Cx43-L + MC3T3-E1: MLO-Y4 cells with low Cx43 expression co-cultured with MC3T3-E1 cells; NC+ MC3T3-E1: MLO-Y4 cell control group co-cultured with MC3T3-E1 cells; MLO-Y4-Cx43-H + MC3T3-E1: MLO-Y4 cells with high Cx43 expression co-cultured with MC3T3-E1 cells. Cx43: green; F-actin: red; DAPI: blue. MLO-Y4 cells and MC3T3-E1 cells are labeled with white arrows). (**B**) The effects of Cx43 expression changes in MLO-Y4 cells on the protein expression of Cx43, β-catenin, Runx2, Osterix, ALP, and OCN in MC3T3-E1 cells in the co-culture model. Data are expressed as mean ± SD, n = 3, * *p* < 0.05. (**C**) The effects of Cx43 expression changes in MLO-Y4 cells on ALP and alizarin red staining in MC3T3-E1 cells in the co-culture model. ALP active site was dark blue or blue-purple and calcium nodules were stained as red. (Cx43-L: MLO-Y4 cells with low Cx43 expression co-cultured with MC3T3-E1 cells; NC: MLO-Y4 cell control group co-cultured with MC3T3-E1 cells; Cx43-H: MLO-Y4 cells with high Cx43 expression co-cultured with MC3T3-E1 cells). The particle analysis tool of Image J was used to calculate the ratio of the stained area, n = 3, * *p* < 0.05.

**Figure 7 ijms-24-10864-f007:**
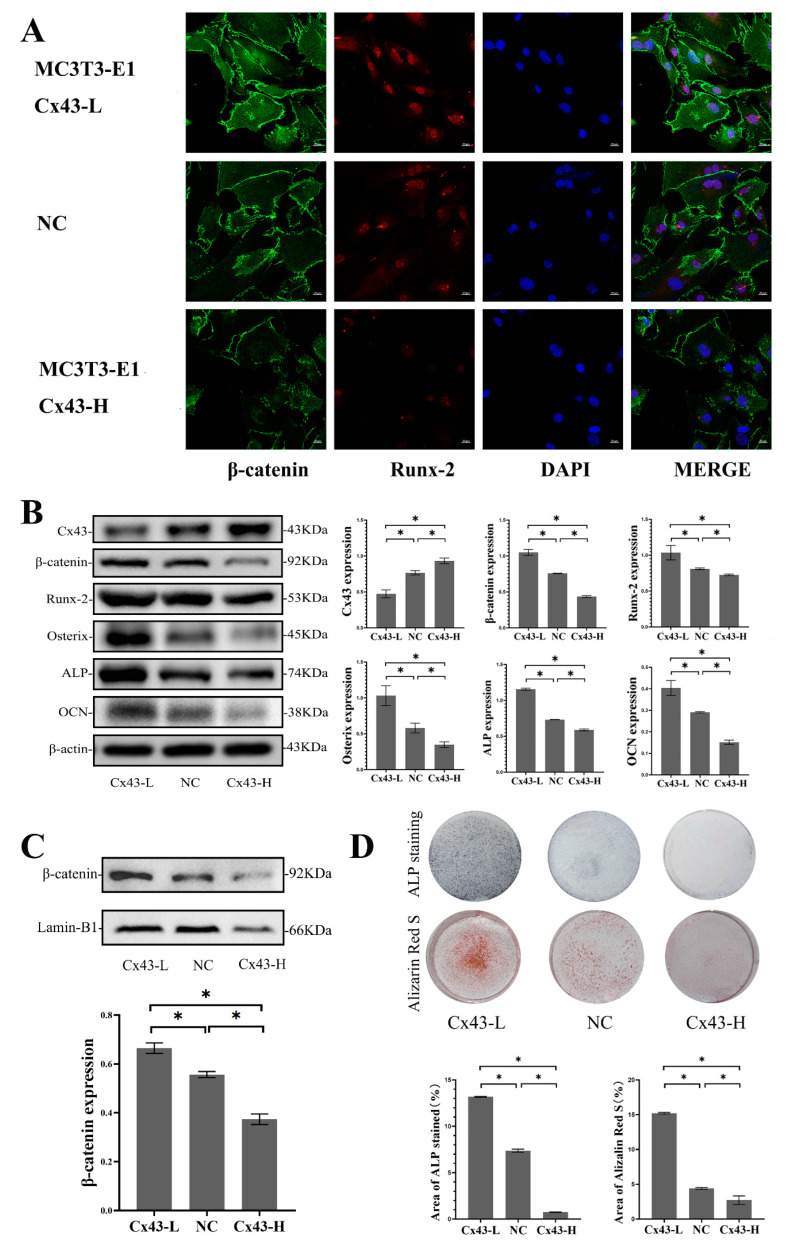
Cx43 reduced osteoblastic differentiation through the inhibition of the Wnt/β-catenin pathway in MC3T3-E1 cells. (**A**) Immunofluorescent staining showing the expression of β-catenin and Runx2 in MC3T3-E1 cells (β-catenin: green; Runx2: red; DAPI: blue. MC3T3-E1-Cx43-L: the group of MC3T3-E1 cells with low Cx43 expression; MC3T3-E1-Cx43-H: the group of MC3T3-E1 cells with high Cx43 expression; NC: negative control group). (**B**) Western blot analysis of the expression of β-catenin, Runx2, Osterix, ALP, and OCN in MC3T3-E1 cells. Data are expressed as mean ± SD, n = 3, * *p* < 0.05. (**C**) Western blot analysis of the β-catenin in the nucleus of MC3T3-E1 cells. Data are expressed as mean ± SD, n = 3, * *p* < 0.05. (**D**) ALP and alizarin red staining and colorimetrical quantitative analysis. ALP active site was dark blue or blue-purple and calcium nodules were stained as red. (Cx43-L: the group of MC3T3-E1 cells with low Cx43 expression; Cx43-H: the group of MC3T3-E1 cells with high Cx43 expression; NC: negative control group). The particle analysis tool of Image J was used to calculate the ratio of the stained area, n = 3, * *p* < 0.05.

**Figure 8 ijms-24-10864-f008:**
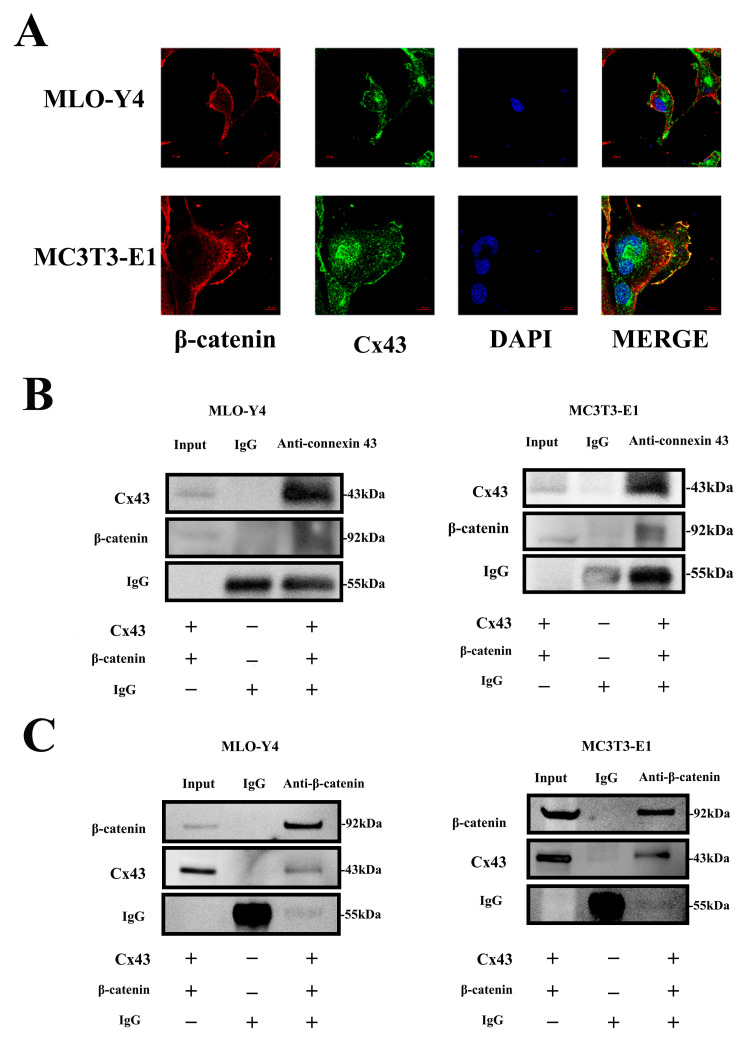
Binding of Cx43 and β-catenin in MLO-Y4 and MC3T3-E1 cells. (**A**) Fluorescence staining showed the co-location of Cx43 and β-catenin in MLO-Y4 and MC3T3-E1 cells mainly along the cell membrane (Cx43: green; β-catenin: red; DAPI: blue). (**B**,**C**) Immunoprecipitation of Cx43 and β-catenin in MLO-Y4 and MC3T3-E1 cells. (Input: total protein supernatant without treatment; IgG: Total protein supernatant supplemented with 10 μg of rabbit IgG; Anti-Cx43: Total protein supernatant supplemented with 10 μg of rabbit-anti-mouse Cx43 antibody; Anti-β-catenin: total protein supernatant supplemented with 10 μg of rabbit-anti-mouse β-catenin antibody).

**Figure 9 ijms-24-10864-f009:**
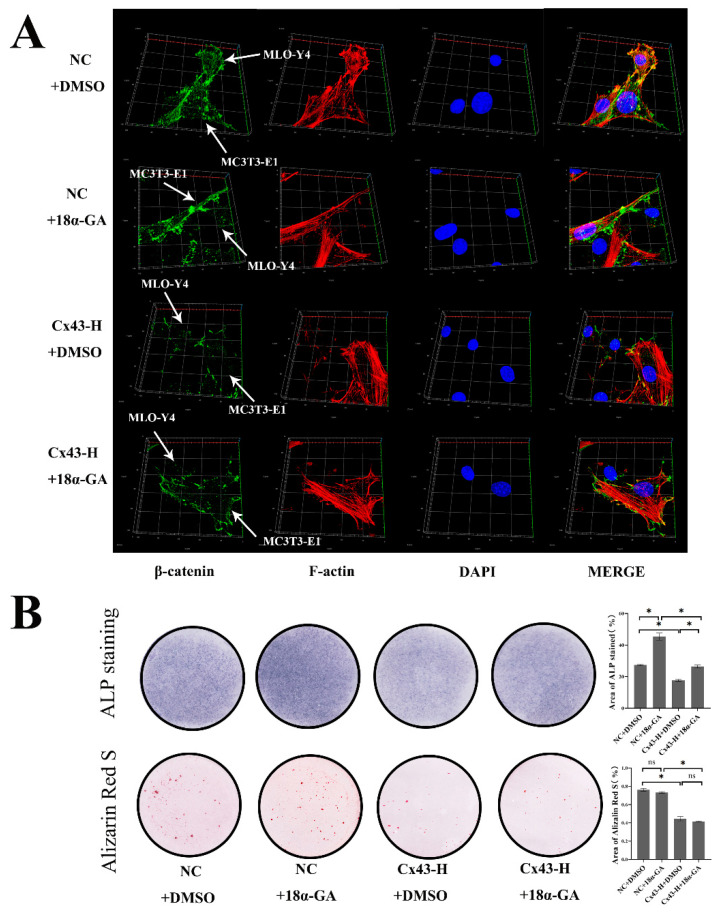
Cx43–GJIC between MLO-Y4 cells and MC3T3-E1 cells decreased β-catenin expression of MC3T3-E1 cells and inhibited early osteoblast differentiation in co-culture models. (**A**) Immunofluorescent staining results after 7 days of co-culture. MLO-Y4 cells and MC3T3-E1 cells are labeled by white arrows. (β-catenin: green; F-actin: red; DAPI: blue). (**B**) ALP and alizarin red staining and colorimetrical quantitative analysis. ALP active site was dark blue or blue-purple and calcium nodules were stained as red. (DMSO: dimethyl sulfoxide; NC+DMSO: MLO-Y4 cells co-cultured with MC3T3-E1 cells with equivalent concentration of DMSO (0.1%); NC+18α-GA: MLO-Y4 cells co-cultured with MC3T3-E1 cells supplementing α-MEM with 10 μM 18α-GA; Cx43-H+DMSO: MLO-Y4 cells with high Cx43 expression co-cultured with MC3T3-E1 cells with equivalent concentration of DMSO (0.1%). Cx43-H+18α-GA: MLO-Y4 cells with high Cx43 expression co-cultured with MC3T3-E1 cells supplementing α-MEM with 10 μM 18α-GA). The particle analysis tool of Image J was used to calculate the ratio of the stained area, n = 3, * *p* < 0.05.

**Figure 10 ijms-24-10864-f010:**
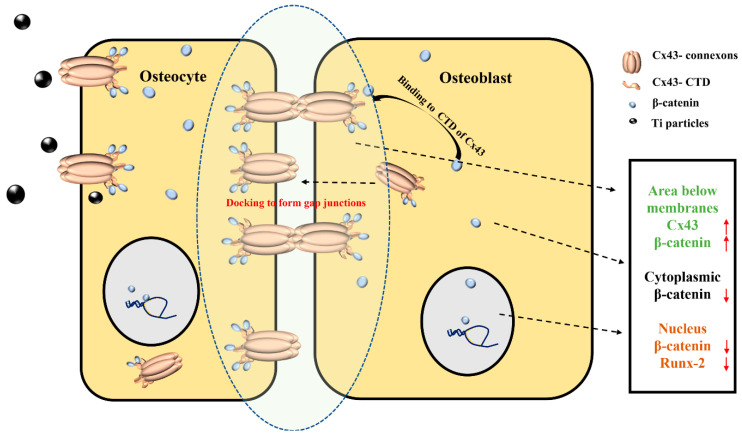
Schematic illustration of the mechanism of osteocyte regulation on osteoblast differentiation via connexin 43/β-catenin pathway. Cx43–connexons between adjacent cell membranes could dock against each other to form a gap junction. After Ti (black) treatment, Cx43 expression of osteocyte cells increased. Osteoblast cells expressed more Cx43–connexons at the membranes to maintain gap junctions’ integrity. The long cytosolic C-terminus (CTD) of Cx43 (orange) could bind β-catenin (blue) and restrict β-catenin nuclear translocation. Only a few β-catenin were translocated into the nucleus. Osteogenic differentiation was inhibited. Upward and downward red arrows represent increased and decreased expression levels, respectively.

**Figure 11 ijms-24-10864-f011:**
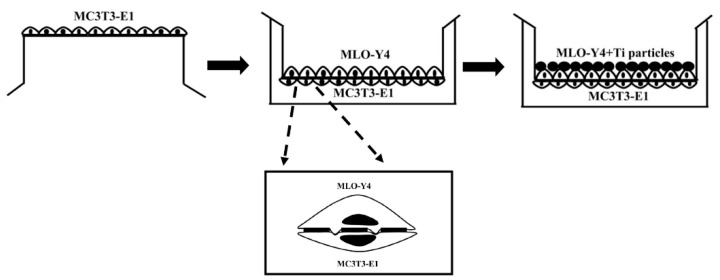
Schematic diagram showing seeding of the two cell types in the co-culture system.

## Data Availability

The datasets generated and/or analyzed during the current study are not publicly available but are available from the corresponding author on reasonable request.

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
