# Peer review of "Osteocytes Exposed to Titanium Particles Inhibit Osteoblastic Cell Differentiation via Connexin 43"

_ijms, 2023, doi:10.3390/ijms241310864_

Round 1

Reviewer 1 Report (New Reviewer)

Summary:

In the manuscript, the authors looked into the effect of Cx43 on titanium particle induced osteolysis with an in vivo Cx43 knockout mice model as well as an ex vivo co-colture system of MLO-Y4 cells and MC3T3-E1 cells. Based on their results, ß-catenin was found to be negatively expresses with Cx43 expression level in both osteocytes and osteoblasts. Inhibition of Cx43 in osteocytes will also reduce the expression of Cx43 in osteoblasts, and leading to an elevated expression of ß-catenin and osteoblast activities. Although, analysis on the osteoclasts markers is completely missing from the study, the current result is adequate. The paper should address the following comments before publication.

Major comments:

1. The Titanium particle size is missing from the method section in 4.3.

2. While the Titanium particles are introduced intraperitoneally around the middle suture of the skull, osteoblasts cell markers are detected in the long bone region. To detect such big systematical effect, microCT images and quantifications of femur should also be provided to support the systemic finding.

Minor comments:

1.     Page 1, line 20, should be “was” not “were”.

2.     Page 1, line 37, “the” should be capitalized.

Author Response

Reviewer 2 Report (New Reviewer)

Hao Chai et al studied titanium particle-induced connexin 43 by in vitro and in vivo models for osteoblast differentiation and claimed that osteocytes exposed to titanium particles inhibited osteoblast differentiation via connexin 43. It is a well-written manuscript with an interesting topic, sufficient data, and various model systems. There are a few things they need to improve further.

1.    The rationale for the link between titanium particles and connexin 43 is somewhat lacking. Are only titanium particles important for connexin 43? Can connexin 43 expressions be regulated in the general bone resorption phenotype or osteogenesis, or does it play a specific role in bone homeostasis?

2.    Is there any potential role of titanium and connexin 43 in osteoclast differentiation and osteoclast activity? It would be better to add or discuss whether CKO mice exhibit a phenotype for osteoclasts or other tissues.

minor

1.    Figure 2B shows some Cx43 expression even in CKO mice. Please explain.

2.    Figure 8C shows no IgG bands in anti-beta catenin panels. Please explain.

Minor editing of English language required.

Author Response

Reviewer 3 Report (New Reviewer)

Chai et al. performed an in vivo and vitro experimental study to the effects of connexin 43 (Cx43) on the regulation of Ti particle-induced osteocyte to osteoblast differentiation. The results revealed that in vivo, the calvarial osteolysis induced by Ti particles was partially attenuated in the CKO mice with  additional findings of increased expressions of β-catenin, Runx2, osterix, ALP, and OCN in CKO femur.  In vitro, Ti particles increased Cx43 expression and decreased β-catenin expression in MLO-Y4 cells and Silencing of Cx43 in MLO-Y4 cells increased the β-catenin expression and over-expressed Cx43 decreased the β-catenin expression. In the co-culture model, Ti treatment of MLO-Y4 cells inhibited the osteoblastic differentiation of MC3T3-E1 cells and Cx43 silencing in MLO-Y4 cells attenuated the inhibitory effects on the osteoblastic differentiation, while the MC3T3-E1 cells displayed decreased Cx43 expression, increased β-catenin expression, activation of Runx2, and promotion of osteoblastic differentiation. Authors concluded that Cx43 expression was found to be negatively correlated to the activity of the Wnt signaling pathway and Ti particle-induced Cx43 elevation in osteocytes may participate in the regulation of osteoblast function.

  The results seem to be interesting, however, the manuscript should add some more clear explanations supporting the conclusive remarks proposed in the abstracts and conclusion.

1.  In the Abstract, authors showed increased expressions of β-catenin, Runx2, osterix, ALP, and OCN in CKO femur. However, these findings were not introduced by treatment of Ti particles. In Figure1, data showed not clear differences between WT and CKO in calvariae. Without additional results showing increased expressions of β-catenin, Runx2, osterix, ALP, and OCN in Ti-induced CKO calvariae, it is hard to conclude that in vivo Cx43-CKO attenuates Ti particle-induced calvarial osteolysis in molecular biological aspects. I think this is an essential part showing in vivo impact of Cx43 in osteocytes on Ti particle-induced calvarial osteolysis. It would be much better if authors would show these additional results.

2. In the Abstract, authors finally suggested as “Cx43 expression was found to be negatively correlated to the activity of the Wnt signaling pathway through two ways, one of them is through binding of β-catenin from its translocation into nucleus, another way might be through the function of gap junctional intercellular communication(GJIC) between adjacent two cells”. However, the present data do not seem to sufficiently support the statement.

Further, based on the results, blocking of Cx43-GJIC seems not show the dramatic changes of ALP and mineral deposition in co-culture experiments (Fig.9), it is still illusive whether Cx43 binding of β-catenin in MC3T3-E1 cells from its translocation into nucleus is responsible for diminished osteoblastic activity of MC3T3-E1 cells in co-culture system as proposed in Schematic illustration of Figure 10.   

3. Independent of co-culture results, the results from Cx43-modulated osteoblastic differentiation/activity especially in MC3T3-E1 cells alone (Fig. 7) seem to show a discrepancy from previous studies showing that Cx43 Knock down leads to diminished osteoblastic differentiation. Although authors shortly introduced previous studies and their controversies in Discussion (Page 15 line 431), I believe more introduction of previous works and discussion about the role of Cx43 on osteogenesis and about discrepancy should be described in the Discussion.     

4. In Fig1. I cannot see the clear differences of Osterix-positive cells among groups in the figures. It might be related to poor quality of staining and/or due to a low magnification of the fields. It looks to be revised.

5. In fig. 3, 5-7, data for nuclear catenin in western blot is missing. It would be much clear evidence for authors´ hypothesis shown in Fig.10, if authors would show the different levels of catenin between cytoplasmic and nuclear compartment among groups.

6. In fig. 8B it seems not so big differences in figures among groups, though quantitative analysis showed 2-fold differences between groups.

7. In discussion, "Overall, our present study demonstrated that osteocytes exposed to titanium parti-441 cles inhibited osteoblastic cell differentiation partially via the Cx43 pathway and inhibi-442 tion of osteocytic Cx43 attenuated the osteolysis induced by wear particles. " (Page 15, line 441) 

I think authors did not show a Cx43 pathway but rather via Cx43. 

Minor revision is required.

Round 2

Reviewer 3 Report (New Reviewer)

I think the revised manuscript has been much improved with valid additional data. Authors have also faithfully replied to reviewers´comments and questionaire.

I believe the revised manuscript acceptable for publication in IJMS.

Author Response

Thank you for your guidance of our revised manuscript (ID: ijms-2412624). 

This manuscript is a resubmission of an earlier submission. The following is a list of the peer review reports and author responses from that submission.

Round 1

Reviewer 1 Report

The authors aim to look at the role of osteocytic Cx43 in periprosthetic osteolysis induced by wear particles, by looking at the regulation of osteocyte to osteoblast differentiation in particle-induced (titanium) osteolysis, particularly in relation to the β-catenin pathway.

There is an array of interesting work presented both in vitro and in vivo, however, when taken together, the story feels incomplete and would benefit from further exploration and discussion, particularly in comparison to how this work adds novelty to recently published work from this group and others. Generally, there are major concerns from this reviewer.

A few key points:

1)     For the cKO of Cx43, more detail regarding the use of which kb of DMP1-cre was used is needed

a.     Alongside that, DMP1 has been recently proven to be expressed in other areas such a muscle/brain, how can the authors further demonstrate its specificity? It would be necessary to do some staining to test the specificity given by this reviewer’s record, this is the first time this group is using this mice model.

2)     For figure 2, for the microCT analysis, the ROI used for analysis, though noted in M&M as being “Micro-CT images were taken from a circular region of interest 425 (ROI of 3 mm in diameter) located at the middle of each calvaria”, it is imperative that this is shown in figure 2 to ensure that the anatomical region was the same.

a.     Alongside that for 2B, the sections do not look to be from a similar sagittal area in different groups….

b.     Also, the BMD quantification and calvarial microCT representative image for cKO+Ti seems to not match with respect to other images, again point important to show ROI

3)     There is a lot of literature of work looking at how Cx43 influences osteoblast differentiation, especially on how the Wnt/B-catenin pathway contributes to this, not all that is presented here is in line with the previous literature, and though some mention is made in the discussion towards the end about this, there is not enough evidence to back the discrepancies and how this work is novel, given again the known overlap between Cx43, osteoblast differentiation, and b-catenin.

4)     Given that this work, is heavily geared at exploring the role of Ocy Cx43 in this context, though Gap junctions (GJs) are briefly mentioned, it should be noted Cx43 is not only present as GJs in osteocytes (hemichannels) and doing cKO or shRNA it is likely to affect both forms. All Cx43 staining, (Fig. 4, 7, & 9) shown in this paper is atypical and does not show solely GJs playing the role here. A better experimental layout should be done using specific Cx channel inhibitors/promotors to look at this relationship.

5)     The use of MLO-Y4 cells, in this context of osteolysis, given its low expression of Sost/sclerostin, which highly regulates wnt/b-catenin and there is also a lot of work on how Cx43 is involved in all this by multiple groups which is not discussed. Which is particularly relevant given the focus of this lab group of SOST in recent publications (Zhang ZH, Jia XY, Fang JY, et al. Reduction of SOST gene promotes bone formation through the Wnt/β-catenin signalling pathway and compensates particle-induced osteolysis. J Cell Mol Med. 2020;24(7):4233-4244. doi:10.1111/jcmm.15084 & Chai H, Zhang ZH, Fang JY, She C, Geng C, Xu W. Osteocytic cells exposed to titanium particles increase sclerostin expression and inhibit osteoblastic cell differentiation mostly via direct cell-to-cell contact. J Cell Mol Med. 2022;26(15):4371-4385. doi:10.1111/jcmm.17460), so it is odd that this is not looked at or discussed. In reference to these recent publications along with others (such as: Wang B, Guo H, Geng T, et al. The effect of strontium ranelate on titanium particle-induced periprosthetic osteolysis regulated by WNT/β-catenin signaling in vivo and in vitro. Biosci Rep. 2021;41(1):BSR20203003. doi:10.1042/BSR20203003) again, it is not clear what this publication is adding to the field in terms of novelty. If Cx43 is to be the focus of novelty here, a much more in-depth analysis is needed to demonstrate its involvement.

Reviewer 2 Report

In this paper, the authors investigate the cellular mechanisms that drive loss of bone caused by titanium particles such as can occur around orthopedic implants, specifically investigating the role of connexin 43.

Key findings of their experiments:

1.  A reciprocal relationship in osteocytic cells between CX43 and B-catenin. Ti particles caused bone erosion in vivo and increased CX43 and decreased B-catenin. Modulation of CX43 levels up and down in osteocyte cells gave opposing effects on B-catenin

2. In a co-culture system between osteocyte and osteoblast cells, Ti particles on the osteocytes or directly overexpressing CX43 in the osteocytes raised CX43 in the osteoblasts but decreased B-catenin and osteogenic markers. Knockdown of CX43 in the osteocytes gave the opposite effects in the osteoblasts.

3. Manipulating CX43 levels in osteoblast cultures had reciprocal effects on their B-catenin levels and osteogenic markers.

4. Connexin43 and B-catenin co-localize and co-immunoprecipitate in both osteocyte and osteoblast cells.

Overall this is a nicely done and interesting study, with well-designed experiments and clear writing to present their science.  Some specific strengths are the lovely immunofluorescence data in Figs 7-9 and balanced discussion at lines 360-378 that discusses both supporting literature as well as prior contrary data.

A number of minor points should be addressed prior to publication.

1. The figures need to be re-numbered. Currently the manuscript starts with Figure 2. Figure 1 is near the end of the manuscript in the Methods section.

2. Lines 104-106 in the legend to figure 2 need a little language revision to be more clear that they are looking in the sagittal suture of the mouse calvarium.

3. Their co-culture system using a filter to physically separate osteocytes and osteoblasts is clever and elegant, and they have described it clearly in the Methods section, found at the end of the paper. At lines 172/173 of their Results section they should add a brief description about their co-culture to be more clear to the reader that they aren't just co-culturing the two cells together in a normal dish. Similarly, I would recommend a little better clarity at line 159 that they have prepared the nuclear fraction for this particular western blot experiment.

4. The header of section 2.9 states that CX43 regulates the Wnt pathway through its binding to B-catenin. This statement is too strong and should be revised.

It is a reasonable interpretation of their data, but they have not actually proven that the physical association between CX43 and B-catenin is what causes the alterations to Wnt signaling. As just one possible alternative, perhaps the presence of functional CX43 gap junctions impacts B-catenin turnover by the axis/GSK3/APC complex and the CX43/B-catenin association isn't really all that important.  I'd suggest overexpressing CX43 defective for B-catenin binding and showing that it loses it's effects on downstream Wnt pathway and osteogenic differentiation to more concretely test this claim.